# High prevalence of hepatitis B virus infection among pregnant women attending antenatal care: a cross-sectional study in two hospitals in northern Uganda

Pontius Bayo,[1] Emmanuel Ochola,[1,2] Caroline Oleo,[2,3] Amos Deogratius Mwaka[3]

[1]St. Mary's Hospital Lacor, Gulu, Uganda
[2]Gulu University Medical School, Gulu, Uganda
[3]Makerere University College of Health Sciences, School of Medicine, Kampala, Uganda

**Correspondence to**
Dr Pontius Bayo;
pontiusby@gmail.com

## ABSTRACT

**Objective:** To determine the prevalence of the hepatitis B viral (HBV) infection and hepatitis B e antigen (HBeAg) positivity among pregnant women attending antenatal clinics in two referral hospitals in northern Uganda.

**Design:** Cross-sectional observational study.

**Setting:** Two tertiary hospitals in a postconflict region in a low-income country.

**Participants:** Randomly selected 402 pregnant women attending routine antenatal care in two referral hospitals. Five women withdrew consent for personal reasons. Data were analysed for 397 participants.

**Primary outcome:** Hepatitis B surface antigen (HBsAg) positivity.

**Results:** Of 397 pregnant women aged 13–43 years, 96.2% were married or cohabiting. 47 (11.8%) tested positive for HBsAg; of these, 7 (14.9%) were HBeAg positive. The highest HBsAg positivity rate was seen in women aged 20 years or less (20%) compared with those aged above 20 years (8.7%), aOR=2.54 (95% CI 1.31 to 4.90). However, there was no statistically significant difference between women with positive HBsAg and those with negative tests results with respect to median values of liver enzymes, haemoglobin level, absolute neutrophil counts and white cell counts. HIV positivity, scarification and number of sexual partners were not predictive of HBV positivity.

**Conclusions:** One in eight pregnant women attending antenatal care in the two study hospitals has evidence of hepatitis B infection. A significant number of these mothers are HBeAg positive and may be at increased risk of transmitting hepatitis B infection to their unborn babies. We suggest that all pregnant women attending antenatal care be tested for HBV infection; exposed babies need to receive HBV vaccines at birth.

**Strengths and limitations of this study**

- In this study, we have evaluated the prevalence of a sexually transmitted viral infection, a risk factor for hepatocellular carcinoma in a population exposed to no condom sexual intercourse in a postconflict region with high rates of HIV infection, another surrogate marker for sexually transmitted infections.
- We also investigated the prevalence of the hepatitis B e antigen, a surrogate measure of the risk of vertical transmission of hepatitis B infection. This is important in determining the need for immediate vaccinations of babies after birth.
- Findings from this study may inform policy on routine testing of pregnant women and immunisation of hepatitis B virus (HBV) exposed babies at birth in addition to the current practice of using combined vaccine at 6 weeks.
- The study had some limitations; it was hospital-based and included a selected population of women with exposure to no condom sexual intercourse and therefore at high risk of sexually transmitted infections including HBV and HIV. In addition, we could not demonstrate evidence for chronicity of hepatitis B infections because we did not perform tests for hepatitis B core antibodies and HBV DNA because of logistical reasons.

perinatal period and early childhood.[2] The risk of becoming a chronic hepatitis B infection carrier is 95% for infections acquired during the perinatal period[3] compared with only 5% for those acquired during adulthood.[4] Up to 50% of HBV carriers die of complications including liver cirrhosis and hepatocellular carcinoma.[5]

Pregnant mothers who test positive for both hepatitis B surface antigen (HBsAg) and hepatitis B e antigen (HBeAg) have 70–90% risk of transmitting infection to their

## INTRODUCTION

Four hundred million people in the world are living with chronic hepatitis B virus (HBV) infection.[1] The majority of these individuals acquired the infection during the

BMJ

newborn infants and about 10–40% risk if they test positive for only HBsAg.[5] [6] Therefore, pregnant women should be routinely screened for HBsAg and hepatitis B vaccine administered at birth to the infants whose mothers test positive.[7] [8] However, this is not the practice in Uganda.

The Uganda National Expanded Program on Immunizations (UNEPI) scaled-up childhood immunisations in 2002[9] incorporated the hepatitis B vaccine into a combination vaccine whose first dose is administered at 6 weeks of age. The 6 weeks window both limits the efficacy of the vaccine in the prevention of vertical transmission and also allows for the potential transmission of HBV through close contacts.[7] The most effective method of preventing HBV infection is through immunisation, which offers over 95% protection against the development of chronic infection.[10] Such immunisation should be done at birth for exposed infants. There is no evidence of protection against perinatal transmission if the first dose of vaccine is given more than 7 days after birth.[11]

In Nigeria, the prevalence of HBV infection among pregnant women was 11% with an HbeAg positivity of 33%.[12] In northern Uganda, there is limited knowledge on the prevalence of hepatitis B infection among pregnant women. The civil war in this region between the government of Uganda and the Lord's resistance army from the late 1980s up to 2006 led to the displacement of as many as 1.7 million people from their homes into internally displaced persons camps.[13] These camps were crowded, traditional and social structures were disrupted and sexually transmitted infections (STIs) such as HBV seemed to have increased. The Uganda HIV serobehavioural survey of 2004/2005 estimated the prevalence of hepatitis B in northern Uganda to be between 18.4% and 24.3%, much higher than the national average of 10%,[14] while in a recent community-based study in Gulu municipality the prevalence of HBV in the general population was estimated at 17.6%.[15]

In this study, we report the prevalence of HBV infection among pregnant women attending antenatal care (ANC) at St. Mary's Hospital Lacor (Lacor) and Gulu Regional referral Hospital using the HBsAg test. We also report HBeAg positivity, a surrogate measure of infectivity among those women who tested positive for HBsAg, and describe the factors associated with HBV infection among these women, with possible implications for testing of pregnant mothers, as well as vaccination of HBV-exposed neonates.

## METHODS
### Study design and setting
This was a cross-sectional study at the Lacor and Gulu regional referral Hospitals. The two hospitals are both in Gulu district in northern Uganda. Lacor hospital is 6 km west of Gulu town; it is a 482 bed capacity teaching hospital[16] and a sentinel site for infectious disease surveillance

in northern Uganda, and has a laboratory with the capacity to separate and store frozen plasma. The Lacor Hospital antenatal clinic (ANC) is visited by 50–80 pregnant women per day, Monday through Friday. The Gulu regional referral hospital, on the other hand, is a 250-bed government owned referral facility located in the centre of Gulu town[16]; the antenatal clinic in Gulu hospital is visited by about 40–60 pregnant women every working day.

### Study population
We included pregnant women attending ANC at the two study hospitals from September 2012 until January 2013, whose gestation age was 28 weeks or more confirmed by clinical history and examination or an obstetric ultrasound scan. We excluded women who had emergency conditions requiring urgent intervention. The two hospitals receive a majority of pregnant women from Gulu district; however, some women attend ANC in other private facilities in the town and health centres.

### Sample size and sampling method
We used the Kish Leslie formula (1965) and a prevalence of HBsAg of 30% for sample size determination, to cater for the North-central Uganda prevalence of about 20%[14] and an additional 10% since pregnant women are engaged in unprotected sex, a known risk factor for STIs compared with the general population.[17] To cater for the possible incomplete responses, we added 10% of the calculated sample size; hence, 402 participants were recruited.

### Sampling procedures
Women were sampled on two working days a week in the two study hospitals: Lacor on Wednesdays and Fridays, while in Gulu, sampling was done on Mondays and Thursdays. All eligible pregnant women attending ANC on the study days were verbally informed of the study immediately after routine ANC health education. We used systematic random sampling, selecting every fifth woman on the ANC waiting line.

### Data collection procedures
At each study site, two midwives were trained for 2 days on study procedures, facts on HBV infections and transmissions, counselling, safety issues, sample collection and transportation as well as site testing for HBsAg.

On obtaining written informed consents, a questionnaire was administered to every selected woman to obtain sociodemographic information including maternal age, gestation age, gravidity, occupation, marital status and highest level of education. Other information on risk factors for transmission of HBV, including a history of previous blood transfusions and a history of scarification, was also obtained. The women were then helped to immediately receive care from the clinic staff.

Participants were informed that those who tested positive for HBsAg would be called back to receive results of another test (HBeAg) to be done on their stored blood

samples. They were also counselled about the hepatitis B vaccine that the study would provide to their infants at birth. The plasma-derived hepatitis B vaccine was administered to infants born to HBsAg positive mothers within 12 h of birth as recommended by the WHO.[18] Each vaccine dose (0.5 mL) contained 10 pg of purified HBsAg.

### Laboratory procedures

Trained research assistants provided pretest counselling on HBV and HIV infections. Five millilitres of blood were then drawn by venipuncture from the cubital fossa under aseptic techniques. The blood samples were immediately put into portable cold boxes with ice packs. The research assistants immediately transported samples to the laboratories at study sites to test for HBsAg (ie, at Lacor and Gulu Hospital Laboratories). In the meantime, the women were helped to obtain ANC. Results were collected back by the research assistants who provided post-test counselling and released results to the participants on the same visit day. Blood samples positive for HBsAg from Gulu hospital were transported on the same day to the Lacor hospital laboratory, frozen at $-80°$C and later transferred to MBN Clinical laboratories in the capital Kampala for HBeAg testing.

Testing for HBsAg was done using the Infectious Diseases ELISA kits provided by Savyon Diagnostics Ltd, Ashdod, Israel, which have a sensitivity of 99% and specificity of 96.7%. Testing for HbeAg was done using the Infectious Diseases ELISA—peroxidase conjugated kits, which have 100% sensitivity and 99.9% specificity, and inbuilt quality controls.

Samples from all the participants were tested for HIV, complete blood counts (CBC), liver alanine aminotransferase (ALT), aspartate aminotransferase, and alkaline phosphatase, using an SMAC auto-analyser (Semi Micro Analyzer Computer, Technicon, USA). CBC was done with an automated analyser, Humacount 60[TS]. HIV tests were performed using a rapid assay for HIV antibody testing.

### Data analysis

Data entered in Microsoft Excel was exported to STATA software V.12 for analysis. We described data using proportions, medians and IQR. Association between participant characteristics and HBsAg positivity was assessed using $\chi^2$ test (or Fisher's exact test as appropriate) for categorical predictors, or Wilcoxon rank-sum test for the continuous laboratory parameters which were not normally distributed (tested using the Shapiro-Wilk test). Logistic regression was performed to predictors of HBsAg positivity at the multivariate level. A p value of ≤0.05 was considered statistically significant in all statistical tests.

### Ethical considerations

Each prospective participant received explanation about the study in their language of choice, mostly Acholi, the major Ugandan language spoken in the study region. They were provided with and given 20 min to study the IRC stamped consent forms in the local language and thereafter requested for their informed consent to participate in the study. Questionnaires were administered only after signed or thumb-printed consents. All participants did not pay for tests done, and test results were provided to the women. All infants born to mothers positive for HBsAg received hepatitis B vaccines at the costs of the study team.

## RESULTS

### Study participants

We approached 402 participants (200 from Lacor and 202 from Gulu Hospital). Five mothers withdrew consent; we therefore included 397 participants in the analysis.

The median age of the participants was 24 years (range 13–43 years). Regarding ethnicity, 89% (n=356) of the participants belonged to the Acholi tribe. Up to 96.2% (n=382) of the women were either married or cohabiting; 71.8% (n=285) of the married women were in a monogamous relationship (table 1).

### Prevalence hepatitis B, HIV and HBeAg positivity

The overall prevalence of HBsAg positivity was 11.8%; the prevalence was 12.7% and 10.9% in the Lacor and Gulu Hospitals, respectively (table 2). HBeAg was positive in 7 of the 47 HBsAg positive women (14.9%).

HBsAg-positive mothers were significantly younger than HBsAg-negative mothers (p=0.002; table 2).

The antibody test for HIV infection was positive among 9.3% (n=37) of participants, but there was no statistically significant association between HIV infection status and hepatitis B prevalence, OR 0.89 (CI 0.30 to 2.65, p=0.839).

### Hepatitis B risk factors

Risk factors including the history of scarification, number of sexual partners, history of blood transfusion or polygamy had no statistically significant relationship with HBsAg positivity (table 2). The liver function tests and complete blood cell counts were similar in HBsAg-positive and HBsAg-negative women. A majority of women had haemoglobin concentrations and platelet counts within normal ranges; these counts were not predictive of HBsAg positivity (table 3).

On multivariable analysis, women 20 years of age or younger were 2.5-fold more likely to test positive than those aged above 20 years; aOR 2.54, CI (1.31 to 4.90); p value 0.006 (table 2 footnote).

## DISCUSSION

This study highlights the high prevalence of HBV infection (11.8%) among pregnant women attending ANC in two hospitals in postconflict northern Uganda. Although the prevalence of HBV and HIV infections in this region

**Table 1** Sociodemographic characteristics of 397 antenatal hepatitis B study participants

| Variable | Frequency, N=397 | Per cent |
|---|---|---|
| Age (years) | | |
| ≤20 | 110 | 27.7 |
| >20 | 287 | 72.3 |
| Education | | |
| Informal education | 30 | 7.6 |
| Primary | 191 | 48.0 |
| Secondary | 140 | 35.3 |
| Tertiary | 41 | 9.1 |
| Tribe | | |
| Acholi | 356 | 89.7 |
| Lango | 17 | 4.3 |
| Others* | 24 | 6.0 |
| Occupation | | |
| Peasant | 220 | 55.4 |
| Professional | 44 | 11.1 |
| Other | 133 | 33.5 |
| Marital status | | |
| Single | 15 | 3.8 |
| Married | 160 | 40.3 |
| Cohabiting | 222 | 55.9 |
| Type of marriage | | |
| Monogamous | 285 | 71.8 |
| Polygamous | 112 | 28.2 |
| Parity | | |
| 0 | 93 | 23.4 |
| 1–4 | 250 | 63.0 |
| 5+ | 54 | 13.6 |
| HIV status | | |
| Negative | 360 | 90.7 |
| Positive | 37 | 9.3 |
| Scarification | | |
| Not done | 43 | 10.8 |
| Done | 354 | 89.2 |

*Other tribes include Madi, Baganda, Jalwo, Karimojong, Banyoro.

exceeds those in most other regions of Uganda that have not experienced prolonged civil conflict and internment in camps, no causal relationship between HBV infection and civil conflict can be inferred from these findings from a cross-sectional study. We also found that about 15% of the HBsAg positive mothers were also HBeAg positive. The prevalence of HBV infection was higher among women aged 20 years or younger (20%) compared with the older women (8.7%). HIV infection among the study population was also high (9.3%). However, there was no significant association between HIV infection and HBV infection among the pregnant women included in this study.

The prevalence of HBV infection among pregnant women in this study is consistent with findings from a study in Nigeria of a prevalence of 11%. The prevalence of HBeAg (33%) was, however, higher in the Nigerian study.[12] The majority of people who get HBV infection after the neonatal period tend to clear the virus over time. The natural history of hepatitis B infection follows three phases: immune tolerant, immune active and immune inactive phases. During the immune active phase when the virus is actively replicating and HBV DNA is high, HBeAg becomes positive and the individual is at a higher risk of transmitting the virus. In the immune inactive phase, the individual has cleared the virus and HBsAg from the blood and becomes less or not infectious to others unless they revert to the immune active phase. The liver enzymes, particularly ALT, are normal during the immune tolerant and immune inactive phases. In our study, the liver enzymes were largely within normal ranges and did not vary significantly between HBsAg-positive and HBsAg-negative pregnant mothers. This may mean that most of our mothers were in the immune intolerant or immune inactive phases of their infections. In the Nigerian study where prevalence of HBeAg was up to 33%, it is probable that the mothers were in the immune active phases and could have had recent infections or were reverting from immune inactive to active phases.[19] The finding in this study that 3 in 20 pregnant women with positive HBsAg are also HBeAg positive means that many unborn babies in northern Uganda are at an even higher risk of infection with HBV. The infants of all these HBsAg positive mothers will need immediate vaccination with HBV vaccine on delivery. This is, however, not the practice in Uganda, and that means the risk of infection is not adequately minimised in these infants. Children who contract HBV infections from their mothers are more likely to develop chronic HBV infection and progress to liver complications associated with active HBV infection including cirrhosis and hepatocellular carcinoma.

To demonstrate a need for a specific affirmative programme to reduce the incidence of complications from chronic HBV infections in this community, we discuss our findings in the context of HBV infections in Uganda as a whole. A review of the sentinel surveillance data shows that the prevalence of HBV infections in this study is higher than that among the HIV positive pregnant women (4.9%) in central Uganda[20] and among the HIV infected adult population (5%) in Rakai, south western Uganda.[21] The prevalence of HBV infection of 18–24% in the general population in northern Uganda is in fact higher than in most parts of Uganda, and higher among men than women,[14 22] and so the findings in this study for the pregnant population just mirrors the background female population prevalence in northern Uganda. In this study, the prevalence of HBV infection was higher among the younger women compared with the older women. This is in variance to findings from a study in Mauritania where there was no significant difference in the mean age of pregnant women who were HBsAg positive compared with those who were negative.[23] Our finding is, however, similar to results from the Uganda national serobehavioural survey in 2005 which showed a prevalence of 8.8% in the age group 15–19 years and increments with age[22] and in Rakai where positive HBsAg

**Table 2**  Association between participants' characteristics (sociodemographic and clinical) and hepatitis B surface antigen (HBsAg) positivity

| Variable | Hepatitis B infection (HBsAg result) | | | | |
| --- | --- | --- | --- | --- | --- |
| | N (+ve) | N (−ve) | Per cent (+ve) | Crude OR (CI) | p Value |
| Overall prevalence | 47* | 350 | 11.8 | – | – |
| Age (years) | | | | | |
| ≤20 | 22 | 88 | 20 | 2.62 (1.41 to 4.89) | 0.002† |
| >20 | 25 | 262 | 8.7 | Ref | |
| Education | | | | | |
| Informal | 6 | 24 | 20 | Ref | |
| Primary | 17 | 174 | 8.9 | 0.39 (0.14 to 1.09) | 0.070 |
| Secondary | 20 | 120 | 14.3 | 0.67 (0.24 to 1.83) | 0.432 |
| Tertiary | 4 | 32 | 11.8 | 0.50 (0.13 to 1.97) | 0.322 |
| Marital status | | | | | |
| Not married | 30 | 207 | 12.7 | Reference | |
| Married | 17 | 143 | 10.6 | 0.82 (0.44 to 1.54) | 0.539 |
| Parity | | | | | |
| 0 | 13 | 80 | 14.0 | Reference | |
| 1–4 | 29 | 221 | 11.6 | 0.81 (0.40 to 1.63) | 0.551 |
| 5+ | 5 | 49 | 9.3 | 0.63 (0.21 to 1.87) | 0.403 |
| HIV status | | | | | |
| Negative | 43 | 317 | 11.94 | Reference | |
| Positive | 4 | 33 | 10.81 | 0.89 (0.30 to 2.65) | 0.839 |
| Scarification | | | | | |
| Not done | 5 | 38 | 10.6 | Reference | |
| Done | 42 | 312 | 11.2 | 1.02 (0.38 to 2.74) | 0.934 |
| History of blood transfusion | | | | | |
| No | 44 | 339 | 11.5 | Reference | |
| Yes | 3 | 11 | 21.43 | 2.10 (0.56 to 7.82 | 0.268 |
| Number of sexual partners in the past 2 years | | | | | |
| 1 | 43 | 334 | 11.4 | Reference | |
| 2 | 4 | 16 | 20.0 | 1.93 (0.62 to 6.08) | 0.254 |

*Of the 47 with an HBsAg positive result, 7 (14.9%) were found hepatitis B e Antigen positive.
†At the multivariate level, only age was a significant predictor of HBsAg positivity, at aOR=2.54 (1.31 to 4.90); p value 0.006.

tests reached the highest level at 8% among the age group 20–29 years.[21] The high prevalence of HBV infection among the younger age group in this study and in the general Ugandan population may be related to the relatively high vulnerability of the younger women to STIs.[24] In northern Uganda where people lived in the camps for more than 20 years, it is possible that these young women themselves acquired perinatal HBV infections from their mothers who could have been exposed to sexually transmitted HBV during life in camps. A study by Råssjö et al[17] showed that women were more disposed to STIs despite risky behaviour being more common among males. Biological and social factors including unemployment and little formal education contribute significantly to a higher prevalence of STIs, including hepatitis B, among adolescent girls.

**Table 3**  Association between participants' laboratory test results and HBsAg positivity

| Variable (median, IQR) | HBsAg positive | HBsAg negative | p Value* |
| --- | --- | --- | --- |
| WCC | 6400 (5180–8100) | 6100 (5100–7300) | 0.294 |
| Lymphocyte count | 1700 (1300–2100) | 1575 (1250–1900) | 0.117 |
| Neutrophil count | 3900 (1300–5100) | 3900 (3040–4900) | 0.619 |
| Haemoglobin | 12.2 (11.4–12.7) | 11.9 (11.1–12.7) | 0.357 |
| ALP | 284 (190–343) | 259 (193–336) | 0.452 |
| AST | 25 (17–27) | 21 (13–29) | 0.251 |
| ALT | 21 (12–30) | 18 (11–26) | 0.314 |

*The Mann-Whitney test was used to test differences in the groups as all the variables were not normally distributed as tested by the Shapiro-Wilk test.
ALT, alanine aminotransferase; ALP, alkaline phosphatase; AST, aspartate aminotransferase; HBsAg, hepatitis B surface antigen; WCC, white cell count.

However, in our study, there were no significant differences in employment status, education levels, marital status and number of sexual partners in the previous 2 years among HBsAg positive participants and those who were negative.

The risk of vertical transmission of HBV infection to the unborn child may be related to the effect of HBV infection on the mother, how she responds to the infection, the timing of the infection with respect to the current pregnancy and the immune status of the mother as well as the levels of HBV DNA.[25] The probability of vertical infection is, however, much increased when the mother is also positive for HBeAg.[3] In one study, vertical transmission was seen in 65% of babies born to mothers who were positive for HBeAg compared with 9.1% for babies born to mothers who were negative for HBeAg.[26] In Senegal, out of 21 infants born to HBsAg positive mothers, 11 were HBsAg positive at birth; and at 6–7 months, five of these were still strongly HBsAg positive and developed antibodies to HBsAg, HBcAg or HBeAg.[27] At present, pregnant women in Uganda are not routinely screened for HBsAg, and the exposed newborns are not immunised at birth against HBV infection. This high prevalence rate of HBsAg positivity among asymptomatic pregnant women in our study shows that there are many infants born who are at high risk of becoming chronic hepatitis B carriers and dying of chronic liver disease at a young adult age in the future.

This study is not without limitations. We did not test for hepatitis B core antibodies (anti-HBc) and HBV DNA; and so there could have been HBsAg negative individuals with isolated anti-HBc and occult HBV infection. However, a recent study among HIV infected pregnant women showed that pregnant women with isolated anti-HBc and occult HBV infection have very low HBV DNA levels and are thus at very low risk to transmit HBV to their infants.[28] We also did not perform high resolution abdominal ultrasound scans;nor did we carry out serial liver enzyme tests to determine which mothers had active hepatitis B infections and may require treatment themselves. However, we referred every mother who tested positive for HBsAg to a competent physician for consultation.

## IMPLICATIONS AND RECOMMENDATIONS

Government and development partners in health need to pay special attention to the high prevalence of infection in this region in order to reduce the cost of care of chronic liver diseases including hepatocellular carcinoma in the future. There is a need to urgently introduce routine screening for HBV infection during pregnancy and provide vaccination at birth for the exposed infants in order to reduce incidences of perinatal infections with HBV.

We recommend further studies to better characterise the pattern of HBV infections among the younger age group (18–25 years). Results of such studies might provide guidance on appropriate methods of interventions to reduce the incidence and prevalence of HBV among these younger populations.

## CONCLUSIONS

There is a high prevalence of hepatitis B infection among pregnant women attending ANC in Gulu and Lacor Hospitals. A high proportion of the HBsAg positive mothers are also HBeAg positive and may be at an increased risk of transmitting HBV infection to their unborn babies. These babies are at high risk of becoming chronic carriers of HBV infections and subsequently increasing the population pool of the virus.

**Acknowledgements** The authors are grateful to the participants for their participation and co-operation. The authors are indebted to the laboratory staff, particularly Ms Peace Amito who prepared the standard operating procedures for the laboratory tests. The authors are also thankful to the research assistants who diligently counselled participants and collected data. This work would have been incomplete without the cooperation of the administrators of the two study hospitals to whom the authors are grateful.

**Contributors** PB, EO and ADM participated in study design and drafting of the manuscript. CO participated in data collection. PB and EO analysed the data. ADM edited and reviewed the final version of the manuscript for important intellectual content and consistency. All authors read and approved the final manuscript.

**Funding** This work was supported by Training Health Researchers into Vocational Excellence in East Africa (THRiVE), grant number 087540 funded by the Wellcome Trust.

**Competing interests** None.

**Ethics approval** Lacor hospital Ethics Review Committees (IRC), and the Uganda National Council of Science and Technology (UNCST), with permission to consider pregnant mothers under 18 years as emancipated minors capable of consenting.

**Provenance and peer review** Not commissioned; externally peer reviewed.

**Data sharing statement** Extra data can be accessed via the Dryad data repository at http://datadryad.org/ with the doi:10.5061/dryad.s1h66.

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
