## [Reviewer comments · BMJ Open]

This paper was submitted to the JECH but declined for publication following peer review. The authors addressed the reviewers' comments and submitted the revised paper to BMJ Open. The paper was subsequently accepted for publication at BMJ Open.

ARTICLE DETAILS

TITLE (PROVISIONAL)	High prevalence of Hepatitis B virus infection among pregnant women attending antenatal care: a cross-sectional study in two hospitals in northern Uganda
AUTHORS	Bayo, Pontius; Ochola, Emmanuel; Oleo, Carolyne; Mwaka, Amos

VERSION 1 - REVIEW

REVIEWER	Frederick Morfaw Department of Obstetrics and Gynaecology, University of Yaounde 1, Cameroon I have no competing interests
REVIEW RETURNED	17-Jul-2014

GENERAL COMMENTS	1. In the abstract section, the authors fail to provide a short background of their study before stating their objectives.2. In the abstract, the authors fail to mention results related to the socio-demographic variables and hepatitis B infection, but this forms a significant part of their results.3. Strengths and limitations are not part of the abstract and should be removed from here.4. Keywords are needed in the abstract section.5. In the methods section, the authors provided no reference for the 21% prevalence which they used to calculate their sample size. In addition, it is not quite clear why an "additional 10% prevalence to account for variations due to the fact that pregnant women are a selected population..." was added.6. The authors failed to define any clear inclusion criteria in the sampling procedures.7. In the results section, table 1 page 11, the column on frequency should be removed as the n=397 and the various percentages are stated.8. Still on table 1 page 11, under type of marriage, 'monogamous' is stated twice hence this result is unclear.9. In the result section, page 12, the authors state that there was no statistically significant difference in Hepatitis B prevalence by HIV status. The authors failed to show how this was calculated. It had to be calculated among the HBsAg(+) and HBsAg(-) and odds ratios with associated p-values calculated.10. In the results section, page 13, table 2 is fundamentally flawed. This is because the authors did tests for association
---

between socio-demographic and clinical characteristics between within the HBsAg(+) group only and used it to draw conclusions on the entire study population. This is inappropriate as it does not tell us the association of these variables with the risk of being HBsAg positive or negative. Tests for associations for these variables (socio-demographic and clinical) should be between the HBsAg (+) and HBsAg(-) groups to be meaningful. If this is done, the results of the multivariate analyses would also change.

11. In the results section, page 14, table 3, it is unclear what statistical test was used to calculate the stated p values?? mean difference? chi-square test?? etc. the statistical test should be clear. in addition, the title of this table should reflect the association the authors are trying to show.

12. In the result section, page 15, the conclusion on the multivariate analyses is wrong as it is based only on analyses within the HBsAg(+) group. Please see comment 10 above and adjust accordingly.

1. In the discussion section, page 16, line 52, it is unclear whether the authors are recommending vaccination for infants born to HBeAg(+) mothers are all HBsAg(+) mothers. The message should be clear as vaccination at birth is recommended for children born to HBsAg(+) mothers.
2. In the discussion section page 17, lines 22-27, the stated prevalence of HBV in the general population is 18-24%, and this is much higher than the 11% among pregnant women in the study. Please give possible explanations implications for this difference. Should more resources not be diverted therefore to fight the infection in the general population in this region?
3. Page 17, line 32, the authors should not mix up the use of incidence and prevalence.
1. Given the limitation pointed out in the testing of association in the result section, you need to re-analyse your data and present the appropriate results. this will equally alter your discussions relative to those results hence a major revision is needed.
 2. In the discussion section the authors spend much time discussing the results of other studies. I suggest they discuss more of their own results and its possible implications within their context.
 3. Page 18, lines 13-27, i do not think the study of Rassjo et al is of any relevance to your study.
 4. The authors fail to make any mention of the use of Hepatitis B immunoglobulin (HBIG) which when associated with to vaccination at birth further minimises the chances of transmission of HBV from mother to child to less than 5%. Could it be made available in this region as well??
 5. Did the authors consider breastfeeding options for HBsAg(+) women??
 6. Page 19 line 40, the authors recommend screening for adolescent girls for HBV, why?? this is not backed by any findings in their study.

2.

REVIEWER	Benson R. Kidenya Catholic University of Health and Allied Sciences Tanzania
REVIEW RETURNED	23-Jul-2014

GENERAL COMMENTS	Yes and I have performed this review The manuscript carries the important finding especially for places where the ante natal services have been disorganized due to civil conflicts and wars. Introduction It is well written, but I advise authors to wind up their introduction a sentence that could emphasize on the significance of the findings from their study. Methods theroutine should be written as the routine How was the 2-day training conducted, and how the assessment was done to ensure that the knowledge and skills required have been grasped? Plasma-derived hepatitis B vaccine was administered to infants born to HBsAg positive mothers within 12 hours of birth. Each vaccine dose (0.5 ml) contained 10 pg of purified HBsAg. Since in Uganda this vaccine is not provided authors give a reference for this dose. Authors should a text such as according to WHO, or as recommend by WHO. Laboratory procedures Trained research assistants provided pre-test counseling on hepatitis B viral and HIV infections. Were these the midwives?, if not how many were they? and where they trained during this study and is so for how long?, and how were they assed? For each of the test the author should specify the manufacturer of that test kit/machine, Country and give the reference where that test has been used previously. Data analysis Authors should state how data was entered from questionnaires/laboratory forms to the computer before being imported to STATA for analysis. Were the data entered straight away from questionnaires to STATA? This phrase "Association between demographic variables and laboratory parameters were assessed using chi-square;" should be amended to read "Association between patients characterists (socio-
---

demographic variables and laboratory parameters) and HBsAg positivity was assessed using Chi-square or Fisher's Exact tests where appropriate;" Chi-square test is NOT appropriate where one of cell has members less than 5, as it was for education, HIV status and number of sexual partners.

Results

When reporting the median it is appropriate to use the interquartile range (IQR) than the using range.

When reporting the proportion/percent/prevalence it is appropriate to use the number of participants as well. For example it is appropriate to report this way "Regarding ethnicity, 89% (356/397) of the participants were of Acholi tribe".

In table 1 the type of marriage monogamous is repeating, I think one should be polygamous.

Education category "None" it not appropriate to say these women have none education, because you will find some of them can count, read, can buy things and knows the change (balance). so they have education but it is informal one. So I advice authors to change this category call it no formal education or informal education (others primary education, secondary educations).

Table 2 title should read association between patients characteristics (socio-demographic and clinical) and HBsAg positivity.

The footnote: **At Multivariate level, only age remained significant predictor of HBsAg positivity, at aOR=2.54 (1.31-4.90); p value 0.006. This statement should amended to read "***At Multivariate level, only age was significant predictor of HBsAg positivity, at aOR=2.54 (1.31-4.90); p value 0.006". If there were other significant factors at the begining (crude OR analysis) the word remaining could have brought sense.

Table 3: We use mean if the data have parametric distribution, and the suitable test is the student t-test. If the data are not parametrically (non-parametrically) distributed we use median, and the student t-test is not appropriate for analysis. So advise authors to use shapiro and wilk test to test the distribution of data and then use the appropriate test for analysis.

This applies to age as well.

It should also be documented on the data analysis section how the normality was tested.

Discussion

Well written, but the findings that about 15% of the HBsAg positive mothers were also HBeAg positive was NOT well discussed comparable to previous studies. For example in Nigeria it was

	HBsAg 11% and HBeAg 33%. What could be the attributing factors to this discrepancy? All these should be discussed. ugandaand should read Uganda and
--	--

VERSION 1 – AUTHOR RESPONSE

Reviewer: Morfaw

1 In the abstract section, the authors fail to provide a short background of their study before stating their objectives

We followed the BMJ Open instructions to authors about the abstract structure; also refer to recent BMJ open publications which give no or minimal background

2 In the abstract, the authors fail to mention results related to the socio-demographic variables and hepatitis B infection, but this forms a significant part of their results

A brief description of some socio-demographic characteristics has been added in the results section. We reported socio-demographics factors comprehensively in the body. Pg2 line 12, 13

3 Strengths and limitations are not part of the abstract and should be removed from here

We have moved this section to the next page

4 Keywords are needed in the abstract section.

The keywords include Hepatitis B, pregnancy/antenatal, Northern Uganda, on title page as instructed by journal Pg 1

5 In the methods section, the authors provided no reference for the 21% prevalence which they used to calculate their sample size. In addition, it is not quite clear why an "additional 10% prevalence to account for variations due to the fact that pregnant women are a selected population..." was added. Reference 14 for the 21% has now been brought forward immediately after 20%.

The overall prevalence used for sample size calculation was a robust 30%, to consider about 20% in Northern Uganda population as reported, and an additional 10% since pregnant women are engaged in unprotected sex, a known risk factor for HBV.

6 The authors failed to define any clear inclusion criteria in the sampling procedures

Inclusion and exclusion criteria has been expanded

We included all pregnant women 28 weeks of gestation age or more confirmed by clinical history and

examination or an obstetric ultrasound scan. The IRB allowed us to consider those under 18 years as emancipated minors. Pg7 line 12-17

7 In the results section, table 1 page 11, the column on frequency should be removed as the n=397 and the various percentages are stated

The authors thank the reviewer, but prefer to keep both columns, as seen in many other studies published in BMJopen, (Jorgensen P, Langhammer A, Krokstad S. et al, 2014; Park JJ. et al, 2014; Hope VD, McVeigh J, Marongiu A, et. al, 2013,) and in other studies (Mofulu NJ, Morfaw FL. et al, 2013) Table1, pg12

8 Still on table 1 page 11, under type of marriage, 'monogamous' is stated twice hence this result is unclear

We regret this; we have cross- checked and corrected this error in table 1: the second 'monogamous' is meant to be 'polygamous' Table1, pg12

9 In the result section, page 12, the authors state that there was no statistically significant difference in Hepatitis B prevalence by HIV status. The authors failed to show how this was calculated. It had to be calculated among the HBsAg(+) and HBsAg(-) and odds ratios with associated p-values calculated

The binary outcome is (HBsAg positive or negative). All the Odds Ratios or p values compare independent variables to the two category outcome

The OR and p value for HIV test association with HBsAg positivity have now been included.

We apologise for the earlier lack of clarity.

P 13, line 7-9

10 In the results section, page 13, table 2 is fundamentally flawed. This is because the authors did tests for association between socio-demographic and clinical characteristics between within the HBsAg(+) group only and used it to draw conclusions on the entire study population. This is inappropriate as it does not tell us the association of these variables with the risk of being HBsAg positive or negative. Tests for associations for these variables (socio-demographic and clinical) should be between the HBsAg (+) and HBsAg(-) groups to be meaningful. If this is done, the results of the multivariate analyses would also change

We have rewritten to clarify the fact that all the statistical tests done, and OR as well as p values recorded were to compare two groups, those who are HBsAg positive and those negative. We have also improved the title of Table 2.

Albeit showing only n positive (as used by studies like Hope VD, McVeigh J, Marongiu A, et. al, 2013 in BMJ Open), the comparisons are for HBsAg positives against negatives with respect to the independent variables.

Pg 10, line 4-10,

Table 2 p 14,

Pg 15 lines 3-7

11 In the results section, page 14, table 3, it is unclear what statistical test was used to calculate the stated p values?? mean difference? chi-square test?? etc. the statistical test should be clear. in addition, the title of this table should reflect the association the authors are trying to show

We have reanalysed and clarified in the footnote and methods section that we used the Wilcoxon rank sum test since the variables were not normally distributed.

The title has now been corrected accordingly. Table3 footnote
Pg 15 line 8

12 In the result section, page 15, the conclusion on the multivariate analyses is wrong as it is based only on analyses within the HBsAg (+) group. Please see comment 10 above and adjust accordingly

The authors beg to differ from this opinion given the explanations above, since the analyses were for two groups. We apologise for the previous lack of clarity.

13 In the discussion section, page 16, line 52, it is unclear whether the authors are recommending vaccination for infants born to HBeAg (+) mothers are all HBsAg(+) mothers. The message should be clear as vaccination at birth is recommended for children born to HBsAg(+) mothers.

Recommendation is for all HBsAg positive mothers (not only HbeAg positive), as clarified in the paper

2. In the discussion section page 17, lines 22-27, the stated prevalence of HBV in the general population is 18-24%, and this is much higher than the 11% among pregnant women in the study. Please give possible explanations implications for this difference. Should more resources not be diverted therefore to fight the infection in the general population in this region?

We have now included the fact that in the Ugandan studies, male prevalence have been relatively higher than females, and this study is only among females, hence the prevalence mirrors the female population prevalence.

3. Page 17, line 32, the authors should not mix up the use of incidence and prevalence

Thanks to the reviewer, we have accordingly modified this.

14 1. Given the limitation pointed out in the testing of association in the result section, you need to re-analyse your data and present the appropriate results. this will equally alter your discussions relative to those results hence a major revision is needed.

The authors hope that the clarifications given above now help. Our re-analysis brings forth similar results, except for Table 3 where there are changes and those changes are because we now used a different statistical test.

2. In the discussion section the authors spend much time discussing the results of other studies. I suggest they discuss more of their own results and its possible implications within their context.

We have improved this section, and specifically added implications.

3. Page 18, lines 13-27, i do not think the study of Rassjo et al is of any relevance to your study.

In our study, the HBV infection was higher among the young pregnant women 20years of age and less, Rassjo et al, explain in their study why the young age group in Uganda may be at a higher risk of STIs including HBV and we think their study is relevant in this context

4. The authors fail to make any mention of the use of Hepatitis B immunoglobulin (HBIG) which when associated with to vaccination at birth further minimises the chances of transmission of HBV from mother to child to less than 5%. Could it be made available in this region as well??

We acknowledge the importance of Hepatitis B immunoglobulin (HBIG) in prevention of vertical transmission of HBV, however, this study was not designed to provide evidence on the efficacy of the preventive methods for vertical transmission. For the babies born to HBsAg positive mothers in this study we provided vaccine alone, the study could not afford the high costs of HBIG at the time. In 2006, the World Health Organization (WHO) provided guideline and said that “on operational and cost-effectiveness grounds, universal use of HBIG is not necessary, especially in countries where pregnant women are not screened for HBsAg”.
(Ref: WHO. Preventing mother-to-child transmission of hepatitis B : operational field guidelines for delivery of the birth dose of hepatitis B vaccine. ISBN 92 9061 206 1. 2006)

5. Did the authors consider breastfeeding options for HBsAg(+) women??

We did not consider breastfeeding options. Evidence given below shows that, breast feeding by a HBsAg positive mother does not appear to pose an additional risk for the acquisition of HBV, although HBsAg has been found in breast milk.

References:

Hill JB; Sheffield JS; Kim MJ; Alexander JM; Sercely B; Wendel GD Risk of hepatitis B transmission in breast-fed infants of chronic hepatitis B carriers. *Obstet Gynecol* 2002 Jun;99(6):1049-52.

Beasley RP; Stevens CE; Shiao IS; Meng HC. Evidence against breast-feeding as a mechanism for vertical transmission of hepatitis B. *Lancet* 1975 Oct 18;2(7938):740-1

6. Page 19 line 40, the authors recommend screening for adolescent girls for HBV, why?? this is not backed by any findings in their study.

We agree with the reviewers' observation that there is no evidence from our study to the fact that vaccinating younger girls with higher infection prevalence will help curb down prevalence of HBV. Nor was our study designed to assess for such relationship. We therefore recommend further studies among this younger age group to better characterize pattern of HBV infections among them.

Reviewer: Benson Kidenya

1 The manuscript carries the important finding especially for places where the ante natal services have been disorganized due to civil conflicts and wars

Thanks for the compliments

2 Introduction: It is well written, but I advise authors to wind up their introduction a sentence that could emphasize on the significance of the findings from their study

A statement about significance has been accordingly added Page6 line 14, 15

3.Methods: the routine should be written as the routine

Modification has been made

4. How was the 2-day training conducted, and how the assessment was done to ensure that the

knowledge and skills required have been grasped?

Trained research assistants provided pre-test counseling on hepatitis B viral and HIV infections. Were these the midwives?, if not how many were they? and where they trained during this study and is so for how long?, and how were they assessed?

Two midwives from each centre were trained for two days on study procedures, facts on HBV infections and transmissions, counselling, safety issues, sample collection and transportation as well as site testing for HBsAg. They practiced these study procedures in a pilot sample of patients prior to the study.

5. Plasma-derived hepatitis B vaccine was administered to infants born to HBsAg positive mothers within 12 hours of birth. Each vaccine dose (0.5 ml) contained 10 pg of purified HBsAg. Since in Uganda this vaccine is not provided authors give a reference for this dose. Authors should add a text such as according to WHO, or as recommended by WHO.

We have improved this sentence to reflect this advice

6. Laboratory procedures:

For each of the tests the author should specify the manufacturer of that test kit/machine, Country and give the reference where that test has been used previously

We regret having omitted this point previously. We have improved this detail in the manuscript. HBsAg test was done using kits provided by Savyon, Diagnostics Ltd, Ashdod, Israel, lot No 41104 – p0396100

7. Analysis:

Authors should state how data was entered from questionnaires/laboratory forms to the computer before being imported to STATA for analysis. Were the data entered straight away from questionnaires to STATA?

Data was entered into Microsoft Office Excel 2007, and exported to STATA for analysis

8. This phrase "Association between demographic variables and laboratory parameters were assessed using chi-square;" should be amended to read "Association between patients characteristics (socio-demographic variables and laboratory parameters) and HBsAg positivity was assessed using Chi-square or Fisher's Exact tests where appropriate;" Chi-square test is NOT appropriate where one of the cells has members less than 5, as it was for education, HIV status and number of sexual partners

We have modified the sentence to reflect the recommendation.

Fisher's exact test results were always assessed, and reported where used.

10. Results: When reporting the median it is appropriate to use the interquartile range (IQR) than the using range.

Thanks, this has been corrected, for the lab parameters. We retained range for age.

11. When reporting the proportion/percent/prevalence it is appropriate to use the number of

participants as well. For example it is appropriate to report this way "Regarding ethnicity, 89% (356/397) of the participants were of Acholi tribe".

Thanks, we have improved on this and shown the n, since all proportions are out of 397.

12. In table 1 the type of marriage monogamous is repeating, I think one should be polygamous.

We regret this, we have cross- checked and corrected this error in table 1: the second 'monogamous' is meant to be 'polygamous'

13. Education category "None" it not appropriate to say these women have none education, because you will find some of them can count, read, can buy things and knows the change (balance). so they have education but it is informal one. So I advice authors to change this category call it no formal education or informal education (others primary education, secondary educations).

Thanks for the advice. "None" Education is now rewritten to "Informal" Education

14. Table 2 title should read association between patients characteristics (socio-demographic and clinical) and HBsAg positivity.

We have re-written this.

15. The footnote: **At Multivariate level, only age remained significant predictor of HBsAg positivity, at aOR=2.54 (1.31-4.90); p value 0.006. This statement should amended to read "***At Multivariate level, only age was significant predictor of HBsAg positivity, at aOR=2.54 (1.31-4.90); p value 0.006". If there were other significant factors at the begining (crude OR analysis) the word remaining could have brought sense.

We have also modified this

16. Table 3: We use mean if the data have parametric distribution, and the suitable test is the student t-test. If the data are not parametrically (non-parametrically) distributed we use median, and the student t-test is not appropriate for analysis. So advise authors to use shapiro and wilk test to test the distribution of data and then use the appropriate test for analysis.

This applies to age as well.

It should also be documented on the data analysis section how the normality was tested

Thanks for detecting this oversight. We have tested for normality using Shapiro wilk test, finding them not normally distributed, thus we did Mann Whitney test for comparing groups, and reported Median (IQR), with the ranksum test p values

Age was purposively categorised for better clinical and comparative significance

Tests for normality and significance of the independent variables is included in analysis section.

17. Discussion:

Well written, but the findings that about 15% of the HBsAg positive mothers were also HBeAg positive was NOT well discussed comparable to previous studies. For example in Nigeria it was HBsAg 11% and HBeAg 33%. What could be the attributing factors to this discrepancy? All these should be discussed.

ugandaand should read Uganda and

We have improved the discussion section, more elaboration has now been made on Hep B e antigen, and also attempts to explain differences in prevalence.

Pages 17-20

Thanks

VERSION 2 – REVIEW

REVIEWER	Benson Kidenya Catholic University of Health and Allied Sciences, Mwanza, Tanzania
REVIEW RETURNED	30-Aug-2014

GENERAL COMMENTS	Yes and I have performed this review Thanks for faithfully working on my suggested comments.
---